# Discovery of a potent, selective, and tumor-suppressing antibody antagonist of adenosine A$_{2A}$ receptor

Linya Wang[1], Pankaj Garg[2], Kara Y. Chan[3], Tom Z. Yuan[1], Ana G. Lujan Hernandez[1], Zhen Han[1], Sean M. Peterson[4], Emily Tuscano[5], Crystal Safavi[1], Eric Kwan[1], Mouna Villalta[1], Melina Mathur[1], Joyce Lai[1], Fumiko Axelrod[1], Colby A. Souders[1], Chloe Emery[1], Aaron K. Sato[1]*

1 Twist Bioscience, San Francisco, California, United States of America, 2 Gilead, Foster City, California, United States of America, 3 Slingshot, Los Angeles, California, United States of America, 4 Nurix Therapeutics, San Francisco, California, United States of America, 5 Sartorius, Fremont, California, United States of America

* asato@twistbioscience.com

**Data Availability Statement:** All relevant data are within the paper and its Supporting information files.

## Abstract

New immune checkpoints are emerging in a bid to improve response rates to immunotherapeutic drugs. The adenosine A$_{2A}$ receptor (A$_{2A}$R) has been proposed as a target for immunotherapeutic development due to its participation in immunosuppression of the tumor microenvironment. Blockade of A$_{2A}$R could restore tumor immunity and, consequently, improve patient outcomes. Here, we describe the discovery of a potent, selective, and tumor-suppressing antibody antagonist of human A$_{2A}$R (hA$_{2A}$R) by phage display. We constructed and screened four single-chain variable fragment (scFv) libraries—two synthetic and two immunized—against hA$_{2A}$R and antagonist-stabilized hA$_{2A}$R. After biopanning and ELISA screening, scFv hits were reformatted to human IgG and triaged in a series of cellular binding and functional assays to identify a lead candidate. Lead candidate TB206-001 displayed nanomolar binding of hA$_{2A}$R-overexpressing HEK293 cells; cross-reactivity with mouse and cynomolgus A$_{2A}$R but not human A$_1$, A$_{2B}$, or A$_3$ receptors; functional antagonism of hA$_{2A}$R in hA$_{2A}$R-overexpressing HEK293 cells and peripheral blood mononuclear cells (PBMCs); and tumor-suppressing activity in colon tumor-bearing HuCD34-NCG mice. Given its therapeutic properties, TB206-001 is a good candidate for incorporation into next-generation bispecific immunotherapeutics.

## Introduction

Immunotherapy has transformed the treatment of cancer from its initial focus on tumor-targeted cytotoxic therapies to next-generation therapies targeting the immune system. New therapeutics, including immune checkpoint inhibitors and chimeric antigen receptor (CAR) T cells enjoyed almost immediate clinical success; for example, from 2011 to 2016, more than 60% of eligible patients were able to access lifesaving anti-PD-1 agents within 4 months of FDA approval [1]. Despite intense enthusiasm and rapid uptake, only an

**Funding:** The author(s) received no specific funding for this work.

**Competing interests:** The authors have declared that no competing interests exist.

**Abbreviations:** A$_1$R, adenosine A$_1$ receptor; A$_{2A}$R, adenosine A$_{2A}$ receptor; A$_{2B}$R, adenosine A$_{2B}$ receptor; A$_3$R, adenosine A$_3$ receptor; Ab, antibody; ADCC, antibody-dependent cellular toxicity; ATX-GK, Alloy Therapeutics Gamma-Kappa; cAMP, cyclic AMP; CAR, chimeric antigen receptor; ELISA, enzyme-linked immunosorbent assay; GPCR, G protein-coupled receptor; HEK293, human embryonic kidney 293; IgG, immunoglobulin G; NECA, N-ethylcarboxamidoadenosine; NK, natural killer; PBMC, peripheral blood mononuclear cell; scFv, single-chain variable fragment; TR-FRET, time-resolved fluorescence energy transfer.

estimated 13% of all cancer patients respond to immunotherapy [2]. Moreover, relapse is still a distinct possibility with immunotherapy, as tumors can develop resistance by mutating tumor antigens beyond recognition, impeding the process of antigen presentation, or otherwise interfering with T cell functions [3]. Combination therapies that target multiple immune checkpoints in parallel are key to overcoming resistance to immunotherapy, yet currently approved agents target only two pathways: CTLA-4 and PD-1/PD-L1. Although combining agents targeting these two pathways has proven more effective than monotherapy in clinical trials [4,5], the discovery of agents targeting alternative checkpoints may expand the proportion of cancer patients that respond to immunotherapy and buttress efforts to minimize adverse events and resistance through novel combination therapeutics [3,6].

Inhibition of the G protein-coupled receptor (GPCR) adenosine A$_{2A}$ receptor (A$_{2A}$R) has been proposed as a strategy for preventing immunosuppression by extracellular adenosine in the tumor microenvironment [7]. Dying, hypoxic, starving, or inflamed cells—all of which can be found in solid tumors—release ATP extracellularly, where it has pro-inflammatory and anti-tumor effects. Hypoxia, inflammation, genetic alterations, and cancer cell dedifferentiation upregulate the ectonucleotidases CD39 and CD73 in solid tumors, enabling the conversion of immunostimulatory ATP into immunosuppressive and tumorigenic adenosine at micromolar concentrations. A natural ligand of A$_{2A}$R, extracellular adenosine exerts its pro-tumor effects on a variety of cell types but primarily on tumor-infiltrating antigen-presenting cells and lymphocytes, as demonstrated by the restoration of antitumor immunity after genetic or pharmacological targeting of the extracellular adenosine pathway in preclinical tumor models. In support of this approach, several small molecule inhibitors of A$_{2A}$R are currently being developed for the treatment of a variety of cancers, either alone or in combination with other anticancer agents. Examples include AZD4635 (AstraZeneca), Ciforadenant (Corvus Pharmaceuticals), EOS100850 (iTeos Therapeutics), Etrumadenant (Arcus Biosciences), and Taminadenant (Pablobio/Novartis).

While small molecules possess some advantages over antibodies in the context of cancer treatment—namely, oral bioavailability and better tumor penetrance—small molecule immune checkpoint inhibitors have not yet demonstrated superiority over antibodies in clinical studies. By contrast, the long half life, exquisite specificity, and engineerability of antibodies make them desirable for immunotherapy development, as demonstrated by the commercial and clinical success of currently approved immunotherapeutic antibodies targeting CTLA-4 and the PD-1/PD-L1 pathway. The dominance of small molecules among drugs targeting A$_{2A}$R in development may also reflect a key challenge associated with discovering antibodies against GPCRs: GPCR antigens for antibody discovery are notoriously difficult to generate and solubilize in native conformations [8].

In this paper, we describe the discovery and characterization of antagonistic antibodies targeting A$_{2A}$R. Our anti-A$_{2A}$R antibody discovery pipeline is illustrated in Fig 1. To maximize the success of this discovery campaign, we prepared a number of synthetic and immunized single-chain variable fragment (scFv) phage libraries using Abveris DiversimAb, Abveris DivergimAb, and Alloy Therapeutics Gamma-Kappa (ATX-GK) mice for immunization. Each library was panned against A$_{2A}$R alone or antagonist-stabilized A$_{2A}$R, the latter of which we hypothesized would increase the number of functional antagonist hits identified. This comprehensive discovery campaign ultimately yielded a specific, high-affinity, cross-reactive humanized antibody antagonist of A$_{2A}$R with *in vivo* tumor suppressing activity (TB206-001).

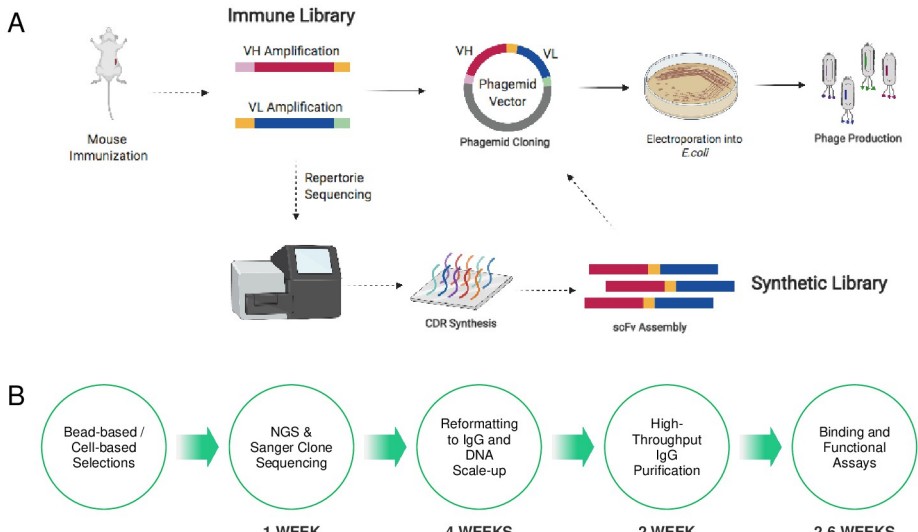

**Fig 1. Accelerated workflow for the discovery of antagonistic A$_{2A}$R antibodies.** (A) Immune and synthetic libraries were sourced and built from antibody diversities obtained from hA$_{2A}$R-immunized Abveris DiversimAb/DivergimAb mice and humanized ATX-GK mice, as described in the materials and methods. (B) Pipeline for the discovery of antagonistic A$_{2A}$R antibodies from immune and synthetic libraries. After phage panning, hits were sequenced, reformatted to IgG, purified, and triaged in a series of binding and functional assays.

## Materials and methods

### Construction of synthetic and immunized anti-A$_{2A}$R antibody phage libraries

Four libraries were generated for screening against human A$_{2A}$R: two immunized libraries and two synthetic libraries (Table 1). For all libraries, antibody diversity was sourced either from Abveris DiversimAb/DivergimAb mice, which are engineered to produce a robust immune response and maximal epitope diversity upon immunization, or humanized ATX-GK mice (Fig 1A). Immunizations were performed by Abveris or Alloy Therapeutics. For each platform, 2 mice were immunized with DNA encoding wild-type hA$_{2A}$R and 2 mice were immunized with DNA encoding mutant hA$_{2A}$R (C144S, S334A, S338A) modified to minimize the impact of the receptor's natural signaling on its expression. After immunization, spleens and lymph nodes were pooled by tissue, dissociated into single-cell suspensions, and stabilized in RNAlater until further processing. Total RNA was extracted using the RNAqueous™ Total RNA

**Table 1. Summary of libraries and triage results for lead identification.**

| ID | Library | Library Size | Panning Target | Reformatted | Flow Cytometry Hits | cAMP Assay Hits | IFN-γ Release Assay Hits |
|---|---|---|---|---|---|---|---|
| TB191 | Alloy Synthetic Library | 7.5 x 10$^9$ | A2a protein | 95 | 3 | 2 | 0 |
| TB192 | Alloy Synthetic Library | | A2a protein + ZM241385 | 95 | 6 | 4 | 1 |
| TB193 | Abveris Synthetic Library | 4.8 x 10$^9$ | A2a protein | 95 | 2 | 2 | 0 |
| TB194 | Abveris Synthetic Library | | A2a protein + ZM241385 | 95 | 10 | 2 | 2 |
| TB205 | Alloy Immune Library | 2.0 x 10$^8$ | A2a protein | 29 | 6 | 2 | 0 |
| TB206 | Alloy Immune Library | | A2a protein + ZM241385 | 10 | 6 | 2 | 3 |
| TB207 | Abveris Immune Library | 2.4 x 10$^7$ | A2a protein | 12 | 2 | 2 | 0 |
| TB208 | Abveris Immune Library | | A2a protein + ZM241385 | 12 | 4 | 0 | 0 |

Isolation Kit (#AM1912), quality controlled using the Qubit™ RNA IQ Assay Kit (#Q33221), and converted to cDNA using the High-Capacity cDNA Reverse Transcription Kit with RNase Inhibitor (#4374966) and the following thermocycling parameters: 10 min at 25˚C, 120 min at 37˚C, 5 min at 85˚C, hold at 4˚C. For the immune libraries, V$_H$, V$_L$, and V$_K$ genes were amplified by overlap extension PCR in three rounds: the first to amplify the genes, the second to add extension overhangs, and the third to link amplified V$_H$ and V$_L$/V$_K$ genes. PCR products were gel-purified in between each round using the QIAquick Gel Extraction Kit. The final PCR products were digested by SfiI for direct cloning into the pADL-22c phagemid vector (immunized libraries). The resulting immune libraries were precipitated, transformed in *E. coli*, purified, and sequenced on a MiSeq sequencer (15% phiX spike-in; MiSeq Reagent Kit v3). OneTaq 2X Master Mix with Standard Buffer (New M0482) was used for PCR.

To build the synthetic libraries, the dominant V$_H$ and V$_L$ genes usage by each mouse immunization platform was determined by next-generation sequencing of V$_H$, V$_L$, and V$_K$ genes; the nearest corresponding human V-genes was then selected as the library framework sequence. The Abveris synthetic library used IGHV3-21 and IGKV1-39 as the framework sequences, and the ATX-GK synthetic library used IGHV1-8 and IGKV1-39 as the framework sequences. Complementarity-determining regions (CDRs) were also sourced from the immunized mice. Given that the diversity of our NGS datasets were much higher than number of unique clones that can be transformed (~$10^{10}$), we reduced the diversity of our NGS datasets by first removing sequence liabilities (cysteines, stop codons, and glycosylation sites), then generating a Levenshtein distance matrix of $n$ x $n$ dimensions, reducing its dimensionality, clustering sequences, and selecting the sequence centers for inclusion. CDRs were shuffled to generate theoretical diversities of 3 x $10^{13}$ sequences within each library. Full-length antibody fragment sequences were assembled from oligo pools synthesized by the Twist Silicon DNA Synthesis Platform. Library sizes are summarized in Table 1.

## Phage panning

Each phage library was biopanned against solubilized, biotinylated A$_{2A}$R (amino acids 2–317; LeadXpro) on streptavidin-coated magnetic beads in the presence or absence of the small-molecule A$_{2A}$R antagonist ZM-241385 (Sigma-Aldrich # 139180-30-6) [9]. Eight point mutations were introduced in the A$_{2A}$R sequence to further stabilize the protein structure: A54L, T88A, R107A, K122A, N154A, L202A, L235A, and V239A. Stabilizing point mutations were identified from Lee et. al, with the exception of S277 remaining wild-type and the addition of the N154A deglycosylation mutation. Phages were selected over 5 rounds of panning and the resulting scFv fragments screened by enzyme-linked immunosorbent assay (ELISA).

## Reformatting, expression, and purification of monoclonal antibodies

Variable heavy chain and light-chain domains were reformatted to IgG1 for DNA back-translation, synthesis, and cloning into mammalian expression vector pTwist CMV BG WPRE Neo utilizing the Twist Bioscience eCommerce portal. Light chain variable domains were reformatted into kappa and lambda frameworks accordingly. Clonal genes were delivered as purified plasmid DNA ready for transient transfection in HEK Expi293 cells (Thermo Fisher Scientific). Antibodies selected for *in vitro* cell-based assay were expressed in 8 mL or 30 mL cultures using the same expression system for larger production. Cultures were grown for 4–5 days, harvested, and purified with Phynexus Protein A resin tips with Cytiva PrsimA resin on the Hamilton Microlab STAR automated liquid-handling systems. Purified antibodies were concentrated using Amicon, 30 kDa cutoff spin filters. For *in vivo* study, antibodies were expressed in 300 mL cultures and were grown to 5 days, harvested and purified using Bio-work TREN 40

pre-packed columns for low endotoxin purification, followed by Cytiva PrismA pre-packed column on the AKTA pure system. Antibodies were eluted with 50 mM citrate buffer, pH 3.0 and neutralized to final concentration of 148 mM HEPES, pH 6.5. The neutralized eluted IgG1 was dialyzed against 1xDPBS, pH 7.40 overnight and changed to fresh cold 1xDPBS, pH 7.4 once in between. CE-SDS was used to determine antibody purity and confirm molecular weight for all scale of expressions. IgGs used for *in vivo* studies were further characterized by size exclusion chromatography with a high-performance liquid chromatography (Thermo Ultimate 3000). Endotoxin levels were tested for all materials used in the animal study (Endo-safe$^{®}$ nexgen-PTSTM Endotoxin Testing, Charles River); the endotoxin level was <5 EU per kg dosing. The total number of reformatted antibodies for each library is listed in Table 1. TB206-001 was also reformatted to IgG4 for comparison to TB206-001 IgG1. The IgG4 was expressed and purified using the same protocol as IgG1.

## Cell lines for *in vitro* assays

*In vitro* assays were performed in ChemiBrite A$_{2A}$R HEK293 cells (Eurofins, #HTS048L); Multiscreen Division-Arrested HEK293T cells expressing either human A$_{2A}$R (Multispan, #DC1428), mouse A$_{2A}$R (Multispan, #DCm1428A), human A$_1$R (Multispan, #DC1427B), human A$_2$B (Multispan, #DC1429), or human A$_3$R (Multispan, #DC1430); Cynomolgus monkey peripheral blood mononuclear cells (PBMCs) (IQ biosciences, #IQB-MnPB102); and human primary cells including PBMCs (Stemcell, #70025), isolated T cell (Stemcell, # 200–0170, # 200–0164), and isolated NK cell (Stemcell, #70036). All HEK293T cell lines were cultured in DMEM/F12 + 10% FBS + 1X NEAA, and human primary cells were cultured in complete RPMI + 10% FBS + 1X GlutaMax + Pen/Strep.

## Flow cytometry assays

Cells were seeded in 96-well plates at 100,000 cells per well and pelleted at 1200rpm for 4 min. Cells were resuspended in blocking solution (100uL PBS+0.5% BSA per well) and incubated for 1 hr at 4˚C. After blocking, cells were pelleted again, resuspended in 100 nM antibody in PBS+0.5% BSA, and incubated for 1 hr at 4˚C. Cells were washed twice with PBS+0.5% BSA and then incubated in secondary antibody (Jackson ImmunoResearch #115-606-071 [anti-mouse] and #109-135-098 [anti-human]) in PBS+0.5% BSA for 30 min at 4˚C. After two more washes, cells were resuspended in PBS+0.5% BSA and run on the Invitrogen Attune or Sartorius iQue3 flow cytometers. The following antibodies were used as positive controls for each adenosine receptor: Anti-Human Adenosine A1 Antibody (Proteintech #20332-I-AP), Anti-Human Adenosine A$_{2A}$R Antibody (R&D #MAB9497), Anti-human Adenosine A$_{2B}$R antibody (Sigma-Aldrich #SAB2500030), Anti-Human Adenosine A$_{2A}$R/A$_{2B}$R Antibody (R&D, #MAB94972), Anti-Human Adenosine A3 Antibody (Invitrogen #PA5-85704), and Anti-Human Adenosine A$_{2A}$R Antibody (which cross-reacts with mouse A$_{2A}$R; Sigma-Aldrich, #05–717).

## cAMP assay

The LANCE cAMP Assay (PerkinElmer, #AD0262E) was used to test the antagonistic activity of antibody hits. In this time-resolved fluorescence energy transfer (TR-FRET) competition assay, agonist stimulation of cells increases cAMP levels, resulting in decreased assay signal. Antibody antagonists block the agonist-stimulated increase in cAMP levels, resulting in increased assay signal. A$_{2A}$R-expressing HEK293T cells were seeded at 10,000 cells per well in a 96 Half Area Well plate (Corning, #3688) and incubated overnight at 37˚C. Cells were then pretreated with titrated anti-A$_{2A}$R antibodies for 1 hr, after which the A$_{2A}$R

agonist NECA was added at a concentration of 0.5 μM NECA. After a 30 min incubation, cells were lysed and incubated in the dark overnight with the LANCE cAMP Assay Detection Mix. TR-FRET signals were measured using a SpectraMax M5 Reader (Molecular Devices).

## Cytokine release assay

The ability of antibody hits to inhibit IFN-γ release from activated T cells in human PBMCs was assayed using the ELISA MAX™ Deluxe Set Human IFN-γ Kit (BioLegend, #430104). Human PBMCs, isolated T cells or isolated NK cells were seeded at 200,000 cells per well in 96-well plates and incubated for 30 min at 37˚C with anti-A$_{2A}$R antibodies, ZM-241385, or AB928 (MedChemExpress, # HY-129393), after which 1, 3, or 10 μM NECA was added for another 30 min. Dynabeads® Human T-Activator CD3/CD28 (Invitrogen #11161D) were then added at 1 μl of beads per 4 x 10$^6$ cells and incubated for 3 d at 37˚C to activate T cells. Finally, culture media was harvested and used directly for the IFN-γ ELISA according to the manufacturer's instructions, and cells were stained with activation markers to determine activation level by flow cytometry.

## HuCD34-NCG-COLO 205 mouse model

TB206-001 was tested for *in vivo* tumor-suppressing activity in HuCD34-NCG mice inoculated with 5 x 10$^6$ COLO 206 cells. The small-molecule A$_{2A}$R antagonist AZD4635 was used as a positive control. Dosing was initiated once tumors had reached an average ~100 mm$^3$. TB206-001, isotype control, or anti-human-PD-1 (pembrolizumab) were administered intraperitoneally at 10 mg/kg once every 3 d for 15 d. AZD4635 was administered orally at 50 mg/kg twice daily for the same time period. Tumor volumes were measured every 3 d to monitor tumor growth. The experiment was terminated 24 d after dose initiation. Mice were weighed and clinically observed throughout the experiment. Mice were defined as moribund and euthanized on the day if one of the following was observed: (1) persistent loss of body weight of >20%, (2) tumors that inhibit normal physiological function such as eating, drinking, and mobility, (3) open, ulcerated wounds that do not form scabs, (4) tumor volume >2000mm$^3$, or (5) clinical observations of prostration, paralysis, seizures and hemorrhages. The studies were operated under the auspices of the Charles River Laboratories IACUC that is based in Worcester, MA.

## Animal ethics statement

Animal study was performed by Charles River as a service. The animal research is operated under the auspices of the Charles River Laboratories IACUC based in Worcester, MA. For euthanasia, a food-grade CO2 gas is used to put the mouse into a state of deep anesthesia and then perform a cervical dislocation to ensure death. The flow rate of CO2 is controlled to deliver the gas a 1 L/minute. Depending on the experiment, 2.5% isoflurane is used for anesthesia and meloxicam (0.5 mg/kg) is used as an analgesic. Mice are checked twice a day for any health issues that may arise during an experiment. Charles River Laboratories have a full-time veterinarian on staff to assist with treatment of the mice for issues such as ulcerated tumors and fight wounds. The standard practices are based on information presented in The Guide for the Care and Use of Laboratory Animals. (https://grants.nih.gov/grants/olaw/guide-for-the-care-and-use-of-laboratory-animals.pdf).

## Results

### Triage strategy for anti-A$_{2A}$R lead identification

We triaged anti-A$_{2A}$R candidates using a series of *in vitro* binding and functional assays. Although all of our initial scFv candidates bound hA$_{2A}$R during biopanning and ELISA screening, only a fraction of these candidates bound hA$_{2A}$R-overexpressing HEK293 cells when reformatted to IgG1 (called "flow cytometry hits"). We then determined whether these flow cytometry hits functionally inhibited cyclic AMP (cAMP) production downstream of hA$_{2A}$R stimulation using a time-resolved fluorescence energy transfer (TR-FRET) cAMP assay. Finally, we tested whether cAMP functional hits could inhibit interferon-ɣ (IFN-ɣ) release downstream of hA$_{2A}$R activation in activated T cells. Qualitatively, the libraries that we panned against ZM241385-stabilized hA$_{2A}$R yielded more hits in each assay than those panned against hA$_{2A}$R alone. Table 1 summarizes the triaging of hits across these assays. We describe the results of binding and functional assays for the top IgG1 lead, TB206-001, which was derived from the Alloy immune library.

### Biophysical and functional characterization of TB206-001

TB206-001 bound hA$_{2A}$R-overexpressing HEK293 cells with an EC$_{50}$ of 5.76 nM, as measured by flow cytometry (Fig 2A). TB206-001 cross-reacted with mA$_{2A}$R but not human A$_1$, A$_{2B}$, or A$_3$ receptors in the same assay (Fig 2B). TB206-001 also cross-reacted with cynomolgus peripheral blood mononuclear cells (PBMC), which include A$_{2A}$R-expressing cells such as T cells, natural killer (NK) cells, dendritic cells, and macrophages (Fig 2C). These data indicate TB206-001 is a high-affinity, specific, and cross-reactive binder of A$_{2A}$R. Raw data is suported in S1 Table.

As a Gαs-coupled receptor, A$_{2A}$R stimulates the production of cAMP via activation of adenylyl cyclase. We therefore evaluated whether TB206-001 can functionally antagonize the stimulation of A$_{2A}$R by N-ethylcarboxamidoadenosine (NECA), a high-affinity small molecule agonist of A$_{2A}$R, using a cAMP competition assay. This assay measures Gαs-coupled receptor activity using FRET between europium-chelate-streptavidin, biotin-cAMP, and a cAMP-specific Alexa Fluor antibody. Stimulation of A$_{2A}$R-overexpressing HEK293 cells with NECA elevates cAMP intracellularly, resulting in saturation of the cAMP-specific Alexa Fluor antibody by free cAMP and abrogating the FRET signal (a FRET signal is only observed in the absence of A$_{2A}$R stimulation). TB206-001 dose-dependently

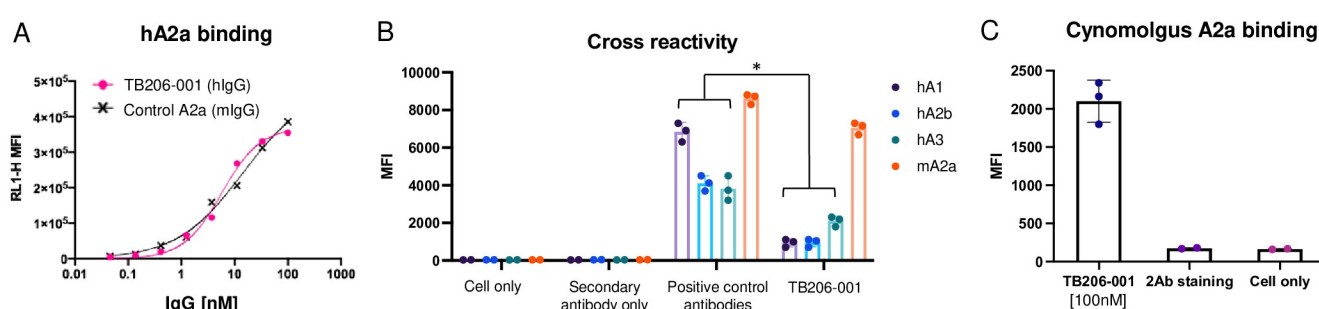

**Fig 2. TB206-001 is a high-affinity, specific, and cross-reactive binder of A$_{2A}$R.** (A) TB206-001 binds to hA$_{2A}$R-overexpressing HEK293 cells with high apparent affinity (EC$_{50}$ = 5.76 nM). (B) TB206-001 cross-reacts with mA$_{2A}$R but not hA$_1$R, hA$_{2B}$R, and hA$_3$R. $p < 0.05$. (C) TB206-001 cross-reacts with cynomolgus PBMCs.

blocked the generation of the FRET signal in $hA_{2A}R$-overexpressing HEK293 cells with an $IC_{50}$ of 3.52 nM (Fig 3A). Thus, TB206-001 is a functional and potent antagonist of $hA_{2A}R$.

Extracellular adenosine inhibits interferon gamma production by T cells, NK cells, and NK T cells by activating protein kinase A (PKA) via the $A_{2A}R$-cAMP pathway [10]. We tested whether TB206-001 could restore T cell production of IFN-γ in NECA-stimulated PBMCs. Treatment of PMBCs with an anti-CD3/CD28 antibody (to activate T cells) resulted in high levels of IFN-γ production that could be abrogated by NECA stimulation. Adding TB206-001 rescued IFN-γ production in T cell-activated/NECA-stimulated PBMCs with three-fold higher potency than the small molecule ZM-241385 ($IC_{50}$ = 6.06 nM vs $IC_{50}$ = 16.75 nM; Fig 3B). Increasing the concentration of NECA decreased the ability of TB206-001 to rescue IFN-γ production in PBMCs (Fig 3C). To distinguish T cell activation from NK cell in PBMC pool, we performed the same stimulation condition as the PBMC assay on isolated T cell and NK cell. After the treatment of TB206-001 or AB928 (A2aR/A2bR small molecule antagonist), isolated T cell showed increased cell proliferation (Fig 3D) and CD25 positive population as a T cell activation marker (Fig 3E). Isolated NK cell demonstrated no effect on IFN-γ secretion (Fig 3F). NK cell can be activated by IL15, but no CD107 activation, an NK cell activation marker, was observed under the same stimulation condition as in PBMC and T cell assay (Fig 3G), showing the IFN-γ secretion in Fig 3B was majorly from T cell activation. In aggregate, our *in vitro* data demonstrate that TB206-001 is a functional and potent competitive antagonist of $hA_{2A}R$ with immunostimulatory effects on T cells. Raw data is supported in S2 Table.

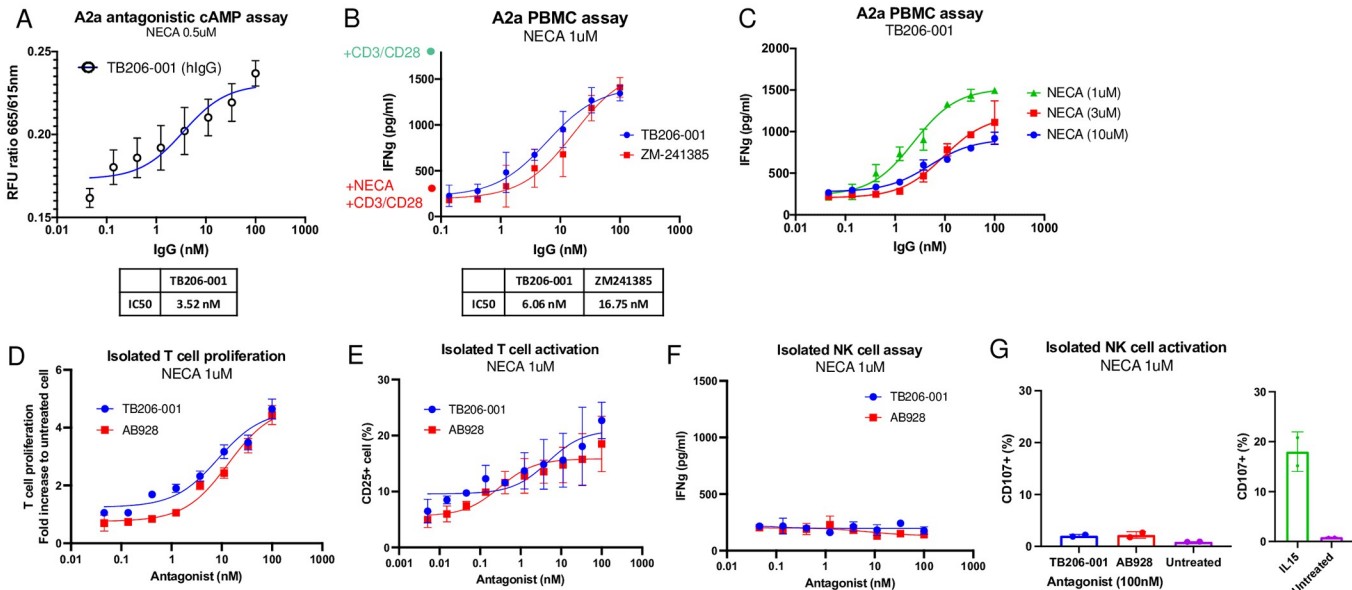

**Fig 3. *In vitro* cAMP cell-based functional assay and primary immune cell activation assay.** (A) Detection of cAMP in HEK293 cells that overexpress $hA_{2A}R$. TB206-001 dose-dependently increased the RFU ratio (665/615 nm), indicating an antagonistic effect on NECA-stimulated cAMP production. (B) TB206-001 antagonized NECA-stimulated IFN-γ release in T cell-activated (CD3/CD28-simulated) PBMCs. The small molecule $A_{2A}R$ antagonist ZM-241385 served as a positive control. (C) Effect of NECA ligand concentration on TB206-001 antagonism of NECA-stimulated IFN-γ release. (D, E) Isolated T cells activation is assessed by cell proliferation and up-regulation of activation marker CD25. The small molecule $A_{2A}R$ antagonist AB928 served as a positive control. (F, G) Isolated NK cell activation is detected by IFN-γ release and activation marker CD107. IL15 stimulation (10 ng/mL) serve as a positive control for NK cell activation.

## Tumor-suppressing activity of TB206-001 in colon tumor-bearing HuCD34-NCG mice

Having established TB206-001 as a functional antagonist of hA$_{2A}$R *in vitro*, we tested whether TB206-001 could suppress tumor growth in a humanized mouse model of human colon cancer. To this end, we compared the tumor-suppressing activity of TB206-001, pembrolizumab, and AZD4635, a clinical-stage small molecule inhibitor A$_{2A}$R [11], in HuCD34-NCG mice inoculated with colon adenocarcinoma (COLO 205) cells (Fig 4A). Like pembrolizumab and AZD4653, TB206-001 suppressed tumor growth in this model (Fig 4B). Because we formatted TB206-001 as an IgG1 for this study, its tumor-suppressing effects could be explained by antibody-dependent cellular toxicity (ADCC), an Fc-mediated function of which the IgG1 subclass is a strong inducer [12]. To rule this out, we reformatted it as an IgG4, which is a poor inducer of Fc-mediated effector functions like ADCC [12], and tested it for A$_{2A}$R binding, A$_{2A}$R antagonism, and *in vivo* tumor suppression. TB206-001-IgG4 bound hA$_{2A}$R, albeit less potently than its IgG1 counterpart (Fig 4C), and primary cell assay (Fig 4D). TB206-001-IgG4 suppressed the growth of COLO 205 tumors in HuCD34-NCG mice with similar potency to TB206-001-IgG1 (Fig 4E). Thus, the tumor-suppressing effects of TB206-001 are not dependent on ADCC. Raw data is supported in S3 Table.

## Discussion

Immunotherapeutics targeting PD-1, PD-L1, and CTLA-4 have transformed cancer care, but only a small minority of patients with cancer respond to these lifesaving treatments. Identifying alternative checkpoints may expand the proportion of patients that respond to immunotherapy. Among the newly identified checkpoints, A$_{2A}$R has been proven noteworthy, with

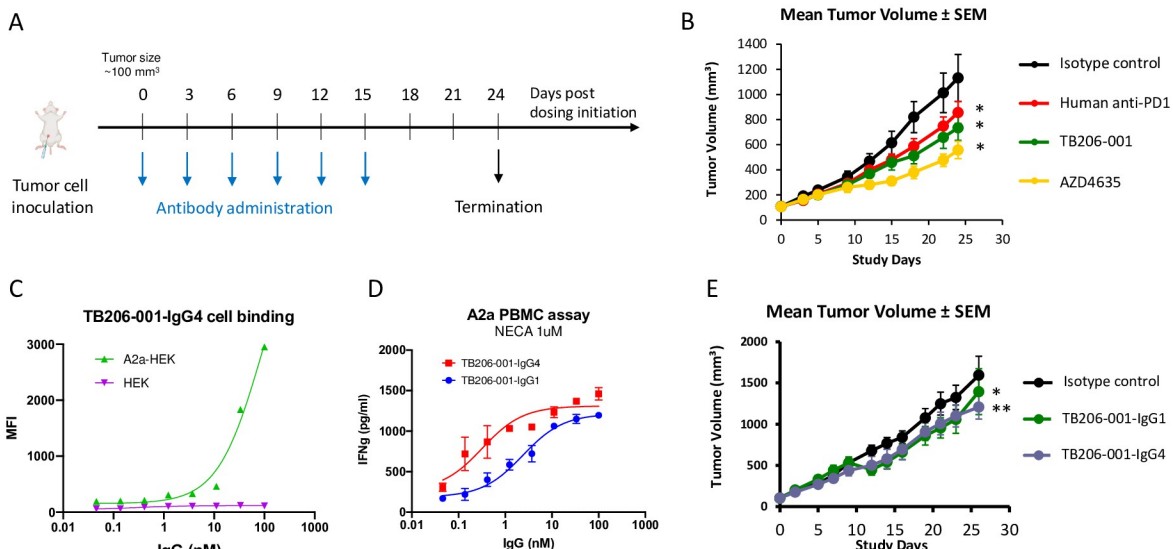

**Fig 4. TB206-001 suppresses the growth of COLO 205 tumors in HuCD34-NCG mice.** (A) Dosing and clinical monitoring schedule. Dosing was initiated when the average tumor volume of the cohort reached ~100 mm$^3$. Mice were monitored three times a week for changes in tumor size and other clinical signs until termination 24 days after dosing initiation. (n = 6) (B) TB206-001-IgG1, pembrolizumab, and the small molecule AZD4635 suppressed the growth of COLO 205 tumors in HuCD34NCG mice. (C) TB206-001-IgG4 binds to hA$_{2A}$R-overexpressing HEK293 cells (EC$_{50}$ = 87.5 nM). (D) TB206-001-IgG4 promotes IFN-γ release from NECA-stimulated, T cell-activated PBMCs. (E) TB206-001-IgG4 and TB206-001-IgG1 suppressed the growth of COLO 205 tumors in HuCD34NCG mice with similar potency. $^*p \leq 0.05$ vs. isotype; $^{**}p \leq 0.01$ vs. isotype.

several small molecule inhibitors of the receptor in the clinical pipeline [13]. Early clinical studies indicate that A$_{2A}$R antagonism is both well-tolerated and effective in multiple indications, either as monotherapy or in combination with existing PD-1/PD-L1-targeting immunotherapeutics [13]. In this work, we discovered TB206-001, a potent, selective, and tumor-suppressing antibody antagonist of hA$_{2A}$R. We contend that inhibiting A$_{2A}$R with TB206-001 can restore immune responses in immunosuppressive tumor microenvironments (S1 Fig).

We speculate that the *in vivo* tumor-suppressing effects of TB206-001 are T cell-dependent because the antibody restored T cell activation following A$_{2A}$R activation *in vitro*. However, other cell types probably contribute to the tumor-suppressing activity of TB206-001, as many other cell types in the tumor microenvironment express A$_{2A}$R, including macrophages, dendritic cells (DCs), NK cells, stromal cells, and tumor cells [10]. Recently, broad immune profiling of MC38-OVA mice treated showed increased levels of NK cells and CD103+ dendritic cells after treatment with the small molecule A$_{2A}$R inhibitor AZD4635 [11], implicating the NK-DC axis in the tumor-suppressing effects of A$_{2A}$R inhibition. A$_{2A}$R inhibition promotes NK cell maturation [14], which may, in turn, promote the maintenance of tumor antigen-presenting dendritic cell populations while suppressing the generation of tolerogenic dendritic cells [15,16]. Myeloid-specific genetic ablation of A$_{2A}$R has also been shown to promote NK and CD8 T cell responses in multiple tumor models [17], potentially implicating tumor-associated macrophages and myeloid-derived suppressor cells in A$_{2A}$R-mediated immunosuppression. Although A$_{2A}$R antagonists may exert their effects on multiple immune cell types, these effects likely converge to promote antitumor T cell activity. In support of this, CD8 T cell depletion abrogated the tumor-suppressing effects of ciforadenant (CPI-444), a potent and selective small molecule inhibitor of A$_{2A}$R, in MC38 tumors [18]. Although we suspect our data reflects an immune mechanism, we cannot rule out a direct effect of TB206-001 on tumor cells in light of evidence showing direct effects of A$_{2A}$R activation on tumor growth and metastasis [19,20]. The effectiveness of A$_{2A}$R antagonists in preclinical tumor models may be explained by the simultaneous targeting of multiple cell types (immune cells and non-immune cells) and biological processes (tumor growth, metastasis, and immune activation).

Small molecule immunotherapeutics have become more popular among drugs in development because they are orally bioavailable, cheaper to develop and produce, and potentially better at penetrating solid tumors [21–23]. Even so, antibodies are generally more specific than small molecules, making them more desirable from a safety perspective. In addition, immunotherapeutic antibodies can be engineered into bispecific antibodies that target multiple checkpoints, improving their potency and decreasing development costs compared to monospecific antibodies. Multiple bispecific antibodies are in development and have shown promising results in preclinical studies, including LY3434172, which targets PD-1 and PD-L1 [24]; LY3415244, which targets TIM-3 and PD-L1 [25]; and YM101, which targets TGF-β and PD-L1 [26]. Our discovery of TB206-001 adds A$_{2A}$R to the immune checkpoint targets available for bispecific development.

A2AR involves in suppressing inflammation and reducing tissue damage, and the regulation of A2AR signaling pathway could be a strategy for tumor therapy. Recent studies have shown that the A2AR is involved in tumor immune escape and is a new immune checkpoint molecule [27,28]. It involves in colorectal cancer with high expressions of A2aR and PD-L1 associated with a poor prognosis [29], gastric cancer with increased A2AR expression associated with TNM stage, lymph node metastasis, distant metastasis and poor prognosis [30], breast cancer with activation of A2AR increased the proliferation and invasion ability [31], and melanoma with increased A2a and A2b level to benefit tumor proliferation [32]. A2aR also plays an important role in the occurrence and development of autoimmune diseases [33]. Activation of A2AR inhibits NF-κB signaling pathway and inflammatory cytokine (IL-1, IL-6,

TNF, IFN) production, and increases the anti-inflammatory cytokine IL-10 release, which leads to autoimmune diseases such as rheumatoid arthritis, systemic lupus erythematosus, or type 1 diabetes [34–36]. Moreover, adenosine-A2AR pathway involves not only in cancer but also Parkinson's disease [37,38]. At present, A2AR antagonists have shown effects in many clinical trials of cancer. However, patients lacking immune cells or the mutation of cancer cell antigen lead to the unrecognition of T cell. The combination of A2AR inhibitors with other therapies such as CAR T can effectively improve the antitumor effect and expand the application of A2AR antagonists. Future combinations treatment with A2a antagonist may become more effective for therapies targeting various diseases [39].

## Supporting information

**S1 Fig. Adenosine pathway targets master checkpoint in tumor microenvironment.** Direct targeting of TB206-001 to A$_{2A}$R can avoid incomplete inhibition of the upstream molecules and promote immune responses.
(TIF)

**S1 Table. Raw data of Fig 2.**
(DOCX)

**S2 Table. Raw data of Fig 3.**
(DOCX)

**S3 Table. Raw data of Fig 4.**
(DOCX)

## Acknowledgments

The authors gratefully acknowledge Alloy Therapeutics for their help developing the Alloy libraries, as well as the Twist Bioscience library team for the design and synthesis of the synthetic libraries.

## Author Contributions

**Conceptualization:** Linya Wang, Pankaj Garg, Melina Mathur, Joyce Lai, Colby A. Souders, Chloe Emery, Aaron K. Sato.

**Data curation:** Linya Wang, Aaron K. Sato.

**Formal analysis:** Linya Wang, Aaron K. Sato.

**Investigation:** Linya Wang, Pankaj Garg, Kara Y. Chan, Tom Z. Yuan, Ana G. Lujan Hernandez, Zhen Han, Sean M. Peterson, Emily Tuscano, Crystal Safavi, Eric Kwan, Mouna Villalta, Fumiko Axelrod, Chloe Emery, Aaron K. Sato.

**Project administration:** Linya Wang, Aaron K. Sato.

**Supervision:** Linya Wang, Fumiko Axelrod, Aaron K. Sato.

**Writing – original draft:** Linya Wang.

**Writing – review & editing:** Linya Wang, Aaron K. Sato.

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
