## [Decision Letter · Decision Letter 0]

18 Dec 2023

PONE-D-23-38500Discovery of a potent, selective, and tumor-suppressing antibody antagonist of adenosine A2A receptorPLOS ONE

Dear Dr. Sato,

Thank you for submitting your manuscript to PLOS ONE. After careful consideration, we feel that it has merit but does not fully meet PLOS ONE’s publication criteria as it currently stands. Therefore, we invite you to submit a revised version of the manuscript that addresses the points raised during the review process.

We look forward to receiving your revised manuscript.

Kind regards,

Alok K Mishra

Academic Editor

PLOS ONE

Journal Requirements:

3. To comply with PLOS ONE submissions requirements, in your Methods section, please provide additional information regarding the experiments involving animals and ensure you have included details on (1) methods of sacrifice, (2) methods of anesthesia and/or analgesia, and (3) efforts to alleviate suffering.

Reviewers' comments:

Reviewer's Responses to Questions

**Comments to the Author**

1. Is the manuscript technically sound, and do the data support the conclusions?

Reviewer #1: Yes

Reviewer #2: Yes

2. Has the statistical analysis been performed appropriately and rigorously? 

Reviewer #1: Yes

Reviewer #2: Yes

3. Have the authors made all data underlying the findings in their manuscript fully available?

Reviewer #1: Yes

Reviewer #2: Yes

4. Is the manuscript presented in an intelligible fashion and written in standard English?

Reviewer #1: Yes

Reviewer #2: Yes

5. Review Comments to the Author

Reviewer #1: The present manuscript describes the development and validation of an antibody antagonist of A2AR. Overall, experiments are designed properly, data are presented clearly, results are summarized well. I recommend that the manuscript be accepted for publication with minor revisions.

Replicates and statistical tests are needed in some experiments. i.e. Figure 2.

Number of mice used in the study needs be provided.

Reviewer #2: In the present manuscript, the author screened and discovered TB206-001 monoclonal antibody antagonist to adenosine A2A receptor, which has a nanomolar binding of hA2AR-overexpressing HEK293 cells and has cross-reactivity with mouse and cynomolgus A2AR. However, it does not cross-react with human A1, A2B, or A3 receptors. In the in vivo model, the antibody has tumor-suppressing activity in colon tumor-bearing HuCD34-NCG mice. The data are exciting and can be a basis for the manuscript: some technical issues of some results and other issues are described below.

1. Although the author has claimed the antagonist activates T-cells in TME. However, natural killer cells also express A2AR and also produce IFN-γ. Does the antagonist affect NK cells?

2. T cell activation could further be assessed by (1) proliferation and (2) up-regulation of activation markers (e.g., IL2RA/CD25).

3. All the bar diagrams used in the study should be modified to dot plots in bar diagrams to increase the transparency related to the independent biological replicate experiments.

4. The authors should discuss more about the immunological aspect of the diseases and cite the relevant manuscripts that would make the manuscript better flow and better suited for publication.

6. PLOS authors have the option to publish the peer review history of their article (what does this mean?). If published, this will include your full peer review and any attached files.

Reviewer #1: No

Reviewer #2: No

---

## [Author Response · Author response to Decision Letter 0]

29 Feb 2024

Answers to reviews’ comments

Reviewer #1: The present manuscript describes the development and validation of an antibody antagonist of A2AR. Overall, experiments are designed properly, data are presented clearly, results are summarized well. I recommend that the manuscript be accepted for publication with minor revisions.

1. Replicates and statistical tests are needed in some experiments. i.e. Figure 2.

2. Number of mice used in the study needs be provided.

Answer 1: Error bars are added to show replicates and statistical difference. Figure 2 is replaced with new graph. 

Answer 2: Number of mice used in the study is 6 mice per group. The description is added in Figure 4 legend. 

Reviewer #2: In the present manuscript, the author screened and discovered TB206-001 monoclonal antibody antagonist to adenosine A2A receptor, which has a nanomolar binding of hA2AR-overexpressing HEK293 cells and has cross-reactivity with mouse and cynomolgus A2AR. However, it does not cross-react with human A1, A2B, or A3 receptors. In the in vivo model, the antibody has tumor-suppressing activity in colon tumor-bearing HuCD34-NCG mice. The data are exciting and can be a basis for the manuscript: some technical issues of some results and other issues are described below.

1. Although the author has claimed the antagonist activates T-cells in TME. However, natural killer cells also express A2AR and also produce IFN-γ. Does the antagonist affect NK cells?

2. T cell activation could further be assessed by (1) proliferation and (2) up-regulation of activation markers (e.g., IL2RA/CD25).

3. All the bar diagrams used in the study should be modified to dot plots in bar diagrams to increase the transparency related to the independent biological replicate experiments.

4. The authors should discuss more about the immunological aspect of the diseases and cite the relevant manuscripts that would make the manuscript better flow and better suited for publication.

Answer 1: NK cell assay is added in Figure 3F and 3G. Utilizing the same stimulating condition and ligand as we use on PBMC assay, the isolated NK cells are not activated by TB206-001 treatment while detecting IFN-γ secretion and CD107 activation marker. 

Answer 2: T cell activation is further assessed by proliferation and up-regulation of CD25 in Figure 3D and 3E. The isolated T cell is activated by TB206-001 treatment. 

Answer 3: All the bar diagrams used in the study is modified to dot plots in bar diagrams to increase the transparency related to the independent biological replicate experiments. i.e. Figure 2. 

Answer 4: The immunological aspect of the various diseases including cancer and autoimmune diseases are added in the discussion session, and the relevant references are cited.

---

## [Decision Letter · Decision Letter 1]

13 Mar 2024

Discovery of a potent, selective, and tumor-suppressing antibody antagonist of adenosine A2A receptor

PONE-D-23-38500R1

Dear Dr. Sato,

We’re pleased to inform you that your manuscript has been judged scientifically suitable for publication and will be formally accepted for publication once it meets all outstanding technical requirements.

Kind regards,

Alok K Mishra

Academic Editor

PLOS ONE

Reviewers' comments:

Reviewer's Responses to Questions

**Comments to the Author**

1. If the authors have adequately addressed your comments raised in a previous round of review and you feel that this manuscript is now acceptable for publication, you may indicate that here to bypass the “Comments to the Author” section, enter your conflict of interest statement in the “Confidential to Editor” section, and submit your "Accept" recommendation.

Reviewer #1: All comments have been addressed

Reviewer #2: All comments have been addressed

2. Is the manuscript technically sound, and do the data support the conclusions?

Reviewer #1: Yes

Reviewer #2: Yes

3. Has the statistical analysis been performed appropriately and rigorously? 

Reviewer #1: Yes

Reviewer #2: Yes

4. Have the authors made all data underlying the findings in their manuscript fully available?

Reviewer #1: Yes

Reviewer #2: Yes

5. Is the manuscript presented in an intelligible fashion and written in standard English?

Reviewer #1: Yes

Reviewer #2: Yes

6. Review Comments to the Author

Reviewer #1: (No Response)

Reviewer #2: In the updated manuscript “Discovery of a potent, selective, and tumor-suppressing antibody antagonist of adenosine A2A receptor”, the authors did address all the previous concerns and now the manuscript is convincing. The authors have addressed all my concerns and have made suggested changes in the manuscript, and I recommend this article for publication.

7. PLOS authors have the option to publish the peer review history of their article (what does this mean?). If published, this will include your full peer review and any attached files.

Reviewer #1: No

Reviewer #2: No

---

## [Editor Report · Acceptance letter]

9 May 2024

PONE-D-23-38500R1 

PLOS ONE

Dear Dr. Sato, 

I'm pleased to inform you that your manuscript has been deemed suitable for publication in PLOS ONE. Congratulations! Your manuscript is now being handed over to our production team.

Kind regards, 

on behalf of

Dr. Alok K Mishra 

Academic Editor

PLOS ONE